

# PCMT1 knockdown attenuates malignant properties by globally regulating transcriptome profiles in triple-negative breast cancer cells

Aili Saiding[1,*], Dilinuer Maimaitiyiming[1,*], Minglan Chen[1], Futian Yan[1], Dong Chen[2], Xinyu Hu[2,3] and Ping Shi[1]

[1] Guangyuan Central Hospital, Guangyuan, China
[2] Center for Genome Analysis, Wuhan Ruixing Biotechnology Co., Ltd., Wuhan, China
[3] Biochemistry & Molecular Biology, Graduate School, Georgetown university, Washington DC, The United States of America
[*] These authors contributed equally to this work.

## ABSTRACT

**Background**. As the most frequently diagnosed cancer in women, Breast cancer has high mortality and metastasis rate, especially triple-negative breast cancer (TNBC). As an oncogene, protein-L-isoaspartate (D-aspartate) O-methyltransferase (PCMT1) is a prognostic biomarker in breast cancer and is highly expressed, while its underlying functions remain unknown.

**Methods**. In this study, we silenced PCTM1 in TNBC MDA-MB-231 cells by short hairpin RNA (shPCMT1) to investigate its cellular functions using cell proliferation, apoptosis, migration, and invasion experiments. Following this, the transcriptome sequencing (RNA-seq) experiment was conducted to explore the molecular targets of PCMT1, including differentially expressed genes (DEGs) and regulated alternative splicing events (RASEs).

**Results**. The results showed that shPCMT1 inhibited the proliferation, migration, and invasion of MDA-MB-231 cells. We obtained 1,084 DEGs and 2,287 RASEs between shPCMT1 and negative control (NC) groups through RNA-seq. The DEGs were significantly enriched in immune or inflammation response and cell adhesion-associated pathways, pathways associated with PCMT1 cellular function in cell migration. The RASE genes were enriched in cell cycle-associated pathways and were associated with the altered cell proliferation rate. We finally validated the changed expression and splicing levels of DEGs and RASEs. We found that 34 RNA binding protein (RBP) genes were dysregulated by shPCMT1, including *NQO1*, *S100A4*, *EEF1A2*, and *RBMS2*. The dysregulated RBP genes could partially explain how PCMT1 regulates the global transcriptome profiles.

**Conclusion**. In conclusion, our study identified the molecular targets of PCMT1 in the TNBC cell line, expands our understanding of the regulatory mechanisms of PCMT1 in cancer progression, and provides novel insights into the progression of TNBC. The identified molecular targets are potential therapeutic targets for future TNBC treatment.

Corresponding authors
Xinyu Hu, 3528879483@qq.com, xh199@georgetown.edu
Ping Shi, 1443197741@qq.com

## INTRODUCTION

Breast cancer is the most commonly diagnosed cancer in women worldwide (*Howard & Olopade, 2021*). Triple-negative breast cancer (TNBC), one subtype of breast cancer, lacks progesterone receptor (PR), estrogen receptor (ER), and human epidermal growth factor receptor 2 (HER2) (*Yin et al., 2020*). It accounts for 10–15% of all breast cancer cases and has strong invasiveness, high rates of metastases and recurrence, and a poor prognosis. Many therapies, such as hormonal or anti-HER2, have small effects on patients with TNBC due to the absence of hormone receptors (HR) and HER2 (*Won & Spruck, 2020*). Thus it is challenging to research TNBC therapy. Chemotherapy remains the standard of care (SOC) for TNBC. However, there is even no SOC for relapsed or refractory TNBC, which indicates the importance of developing a new therapy (*Won & Spruck, 2020*).

Researchers recently found that immune checkpoint inhibitors may facilitate the development of TNBC therapy (*Borri & Granaglia, 2021*). TNBC can also be immunogenic for multiple reasons. For example, the subtypes of TNBC have many genetic mutations and the immune system may treat the aberrant proteins as foreign ones (*Disis & Stanton, 2015*). As more antigens specifically expressed in TNBC cells were discovered, immunotherapy is now considered to be a new promising solution to TNBC (*Li et al., 2018b*). These findings facilitate the use of immunomodulatory agents such as immune checkpoint inhibitors in TNBC clinical trials to further understand immune-related TNBC properties and to find more novel immune markers (*Romero-Cordoba et al., 2019*). However, clinical trials are still necessary to validate the efficacy of molecular targets (*Fabbri, Salvi & Bravaccini, 2020*).

Some proteins are considered to be involved in the regulation of breast cancer, including PCMT1 that is related to the abundance of immune infiltration. A recent study demonstrated that the level of PCMT1 in breast cancer cells is significantly high (*Guo, Du & Li, 2022a*). PCMT1 has been proven to participate in the regulation of the proliferation, apoptosis, and migration of different cancer cells, which further promotes the occurrence and development of cancers (*Dong et al., 2021*). According to a previous study, PCMT1 is a target gene in liver cancer, indicating that it serves as a therapeutic target in multiple cancer treatments (*Amer et al., 2014*). PCMT1 may regulate cancer-related processes such as apoptosis through regulating multiple proteins such as Mst1 (*Yan et al., 2013*). Extracellular matrix (ECM) protein LAMB3 can interact with PCMT1 to activate the integrin-FAK-Src pathway that promotes the adhesion, invasion, and migration of cancer cells. These effects can be reversed by PCMT1-blocking antibody treatment (*Zhang et al., 2022a*; *Zhang et al., 2022b*). PCMT1 can also regulate the stability of regulatory T cells by methylating the promoter of forkhead box P3(FOXP3), which may explain some of its functions (*Ozay et al., 2020*). However, the exact functions and downstream targets of PCMT1 in TNBC are largely unknown.

In this study, we focused on the assumption that PCMT1 may regulate breast cancer-related processes by influencing the global transcriptome profile, including gene expression and alternative splicing events (ASEs). In this study, we knocked down PCMT1 in human TNBC cells (MDA-MB-231). PCMT1-related transcriptome data were obtained using

high-throughput transcriptome sequencing (RNA-seq). The data were then used to analyze the potential targets regulating the transcription and alternative splicing (AS) levels of PCMT1 in MDA-MB-231 cells. The results demonstrate that PCMT1 globally regulates the transcriptome profile in MDA-MB-231 cells, and partly explain the functions on cellular phenotype of PCMT1.

## MATERIALS AND METHODS

### Lentivirus

Sh-PCMT1 lentivirus was purchased from Genephem (Shanghai, China). ShRNA targeting PCMT1 (shPCMT1) sequence was: 5′-GCTAGTAGATGACTCAGTAAA-3′ (sense), while the negative control (NC) sequence was: 5′-TTCTCCGAACGTGTCACGT-3′ (sense).

### Culture and shRNA transfection of cells

The MDA-MB-231 cell line (CL-0150B, Procell, Wuhan, China) were cultured with 5% $CO_2$ at 37 °C in Dulbecco's modified Eagle medium (DMEM) that was supplemented with 10% fetal bovine serum (FBS), 100 µg/mL streptomycin, and 100 U/mL penicillin. MDA-MB-231 cells were infected by lentivirus with MOI = 100. Stable cell lines were obtained by screening with 0.5 ug/ml puromycin and the cells were then harvested for following experiments. The primers for RT-qPCR could be found from Table S1.

### The proliferation assay

The proliferation assay of MDA-MB-231 cells was conducted using a Cell Counting kit-8 (CCK-8, 40203ES76, Yeasen, Shanghai, China). Briefly, stable MDA-MB-231 cell lines were seeded with 10,000 cells/well in 96-well culture plates. Vials without cells were used as blank controls. As the degree of cell polymerization will reach 100% after 96 h or longer, we used 0 h, 24 h, 48 h, and 72 h for assessment. After incubation at 37 °C with 5% $CO_2$, we added 10 µl CCK-8 solution to the culture medium for an additional 3 h at 37 °C. We then measured the optical density (OD) of the cells with a Microplate Reader (ELX800, Biotek, Winooski, VT, USA) at an absorbance of 450 nm. The cell proliferation rate = (experimental OD value − blank OD value)/(control OD value − blank OD value) × 100%.

### Flow cytometric analysis of cell apoptosis

An Annexin V-APC/7-ADD apoptosis detection kit (40304ES60, Yeasen, Shanghai, China) was used to analyze cell apoptosis following the manufacturer's instructions. Stable MDA-MB-231 cell lines were seeded into 6-well plates and cultured for 48 h. The shPCMT1 and NC cells were mixed with 5 µl Annexin V-APC separately and incubated in the dark at room temperature for 5 min, and then mixed with 5 µl 7-AAD reagents and incubated for 5 min, respectively. Then the cells were subjected to flow cytometric (FACSCanto, BD, Franklin Lakes, NJ, USA) analysis to calculate cell apoptosis levels.

### Cell invasion assay

Invasion assays were conducted using transwell chambers (3,422, Corning, Corning, NY, USA). The transwell chambers had 8 µm filter at the bottom and were precoated with a
uniform layer of Matrigel (356234, BD Biosciences, San Jose, CA, USA), diluted for 1:8 using serum-free medium, 100 ul diluted matrigel in chambers was incubated for 1 h at 37 °C with 5% $CO_2$ and removed unsolidified supernatant. Following this, $4 \times 10^5$ cells were added to the inserts. Then the transwell chambers were inserted in medium with 600 ul 10% FBS (10091148, Gibco, Waltham, MA, USA). The detailed steps were conducted following a published study (*Yan et al., 2018*). The invasion cells were calculated under an inverted microscope (MF52-N, Mshot, Guangzhou, China) at 200× magnification.

## Cell migration assay

Migration assay was performed using transwell chambers (3422, Corning, Corning, NY, USA). A total of $2.5 \times 10^5$ cells in 0.2 ml serum-free medium were added to the transwell chambers with 8 $\mu$m filter. Then the chambers were inserted in medium with 600 ul 10% FBS (10091148, Gibco, Waltham, MA, USA). The medium served as a chemoattractant in the lower chamber. The detailed steps were conducted following a published study (*Yan et al., 2018*). The migration cells were calculated under an inverted microscope (MF52-N, Mshot, Guangzhou, China) at 200× magnification. The raw data of apoptosis, proliferation, invasion, and migration results could be found from Table S2.

## RNA extraction and sequencing

The DNAs were removed by RQ1 DNase (Promega, Madison, WI, USA) to extract total RNAs. The quality and quantity of RNAs were then identified by the absorbance at 260 nm/280 nm (A260/A280) using SmartSpec Plus (BioRad, Hercules, CA, USA). For the three biological replicates of shPCMT1 and NC samples, mRNAs were captured from 1 $\mu$g of total RNA by VAHTS mRNA capture Beads (Vazyme, Nanjing, China; N401), and used for directional RNA-seq library preparation by VAHTS Universal V8 RNA-seq Library Prep Kit on Illumina (NR605). The detailed steps were conducted following a published study (*Ren et al., 2023*). Illumina NovaSeq 6000 system was used to generate the raw 150 nt paired-end reads of RNA-seq libraries.

## RNA-seq quality filtering and differentially expressed genes analysis

Raw reads were filtered by removing reads with N bases, trimming adaptors and low-quality bases using a FASTX-Toolkit (Version 0.0.13). Too short reads (<16 nt) were also discarded. The filtered reads were then aligned to the GRCh38 genome by HISAT2 (*Kim, Langmead & Salzberg, 2015*) allowing up to 4 mismatches. Fragments per kilobase of transcript per million fragments mapped (FPKM) was calculated using the uniquely mapped reads (*Trapnell et al., 2010*). The statistical power of this experimental design (three biological replicates), calculated in RNASeqPower (*Hart et al., 2013*) is 1. The R package DESeq2 (*Love, Huber & Anders, 2014*) was used to screen the DEGs. The corrected *p*-value < 0.05 and fold change (FC) > 2 or <0.5 were set as the cut-off criteria to identify DEGs differentially expressed genes differentially expressed genes (DEGs).

## AS analysis

The ABLas pipeline (*Jin et al., 2017*; *Xia et al., 2017*) was used for AS analysis and predicting the regulated alternative splicing events (RASEs) between shPCMT1 and NC samples.

Briefly, ABLas detected nine types of ASEs based on the splicing junction reads, comprising exon skipping (ES), alternative 5′ splice site (A5SS), alternative 3′ splice site (A3SS), mutually exclusive exons (MXE), mutually exclusive 5′UTRs (5pMXE), mutually exclusive 3′UTRs (3pMXE), cassette exon, A3SS&ES, and A5SS&ES. To assess the RASEs by shPCMT1, Student's $t$-test was used to calculate the significance of the ratio alterations of each AS event. Those events with a $P$-value $< 0.05$ were considered as RASEs.

### Reverse transcription and quantitative polymerase chain reaction
Reverse transcription and quantitative polymerase chain reaction (RT-qPCR) was performed to validate the DEGs and RASEs. Detailed steps and data analysis method of this experiment could be found in the published article (*Li et al., 2018a*). The primers for DEGs and RASEs could be found from Table S1. The raw data of RT-qPCR results could be found from Tables S3–S4.

### Functional enrichment analysis
To explore the functional categories of DEGs, KOBAS 2.0 server was used to identify enriched Gene Ontology (GO) and Kyoto Encyclopedia of Genes and Genomes (KEGG) pathways (*Xie et al., 2011*). Hypergeometric test and Benjamini–Hochberg FDR methods were used to define the statistical significance of each pathway.

## RESULTS

### PCMT1 knockdown inhibits the proliferation of MDA-MB-231 cells
A previous study has demonstrated the prognostic value of PCMT1 in breast cancer (*Guo, Du & Li, 2022a*). In this study, we further explored the cellular functions and downstream targets of PCMT1 in MDA-MB-231 cells. The PCMT1-knockdown MDA-MB-231 cell line was constructed by the short hairpin RNA (shRNA) method (shPCMT1); an empty plasmid cell line was used as negative control (NC). According to the RT-qPCR results, the expression levels between NC and shPCMT1 samples were different after transfection (Fig. 1A), consistent with the Western blot result, indicating the successful knockdown of PCMT1 (Figs. 1B, S1A). Based on the TCGA database, the survival curves of PCMT1 in breast invasive carcinoma (BRCA) also indicated that the low PCMT1 expression group had a better prognosis than the high expression group (Fig. 1C), consistent with the previous result. The proliferation experiment also showed that NC samples had a higher proliferation rate than shPCMT1 samples (Fig. 1D). While the apoptosis levels between shPCMT1 and NC samples changed slightly (Figs. S1B–S1C). Taken together, these results demonstrated that silencing PCMT1 decreases the proliferation rate of MDA-MB-231 cells.

### PCMT1 knockdown inhibits the migration and invasion of MDA-MB-231 cells
To further explore the cellular functions of PCMT1 in MDA-MB-231 cells, we conducted cell migration and invasion experiments. The results proved that PCMT1 knockdown

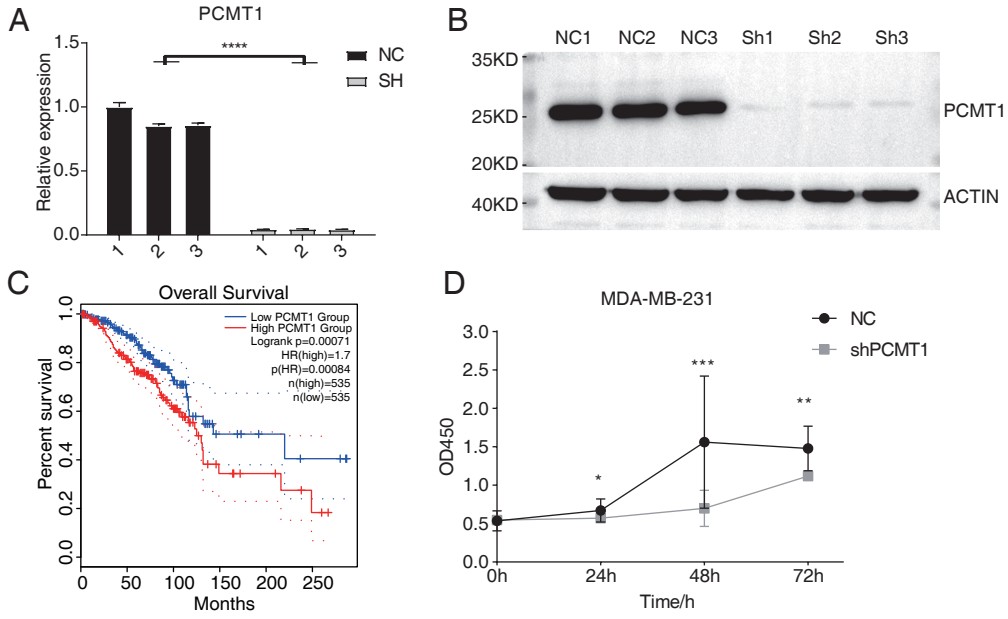

**Figure 1** **PCMT1 knockdown inhibits MDA-MB-231 cell proliferation.** (A) Bar plot shows the RT-qPCR results of NC and shPCMT1 samples. Error bars represent mean ± SEM. $N = 3$. ****$p$-value < 0.0001, Student's $t$-test. (B) The Western blot result shows that the PCMT1 knockdown is successful. (C) The survival curve of BRCA patients by dividing them into two groups according to PCMT1 expression using the GEPIA2 database. (D) The proliferation results of MDA-MB-231 cells after PCMT1 knockdown. Error bars represent mean ± SEM. $N = 3$. *$p$-value < 0.05, **$p$-value < 0.01, ***$p$-value < 0.001, Student's $t$-test.

significantly inhibited the migration and invasion ability of MDA-MB-231 cells (Figs. 2A–2D). These results together indicate that PCMT1 may play an important role in the progression of breast cancer, including TNBC.

## PCMT1 regulates the global gene expression profile in MDA-MB-231 cells

We then used RNA-seq to identify the changed transcriptome profiles by shPCMT1 and the molecular targets of PCMT1. According to the RNA-seq result, the expression level of PCMT1 significantly decreased in shPCMT1 samples (Fig. 3A), confirming the successful inhibition of PCMT1 expression in MDA-MB-231 cells. We then performed principal component analysis (PCA) for all detected genes. The result shows that the shPCMT1 samples were separated from the NC samples by the first component, which accounted for 41.7% of all sample variation (Fig. 3B). To identify the molecular targets of PCMT1, we performed DEG analysis and obtained 1084 DEGs between shPCMT1 and NC samples after PCMT1 knockdown, comprising 771 up-DEGs and 313 down-DEGs (Fig. 3C). The hierarchical clustering heatmap also shows a consistent expression pattern in shPCMT1 and NC samples (Fig. 3D). We then performed a functional enrichment analysis of Gene Ontology (GO) database to explore the underlying functions of DEGs. The up-DEGs were enriched in immune and inflammation-associated pathways and cell adhesion- and cell

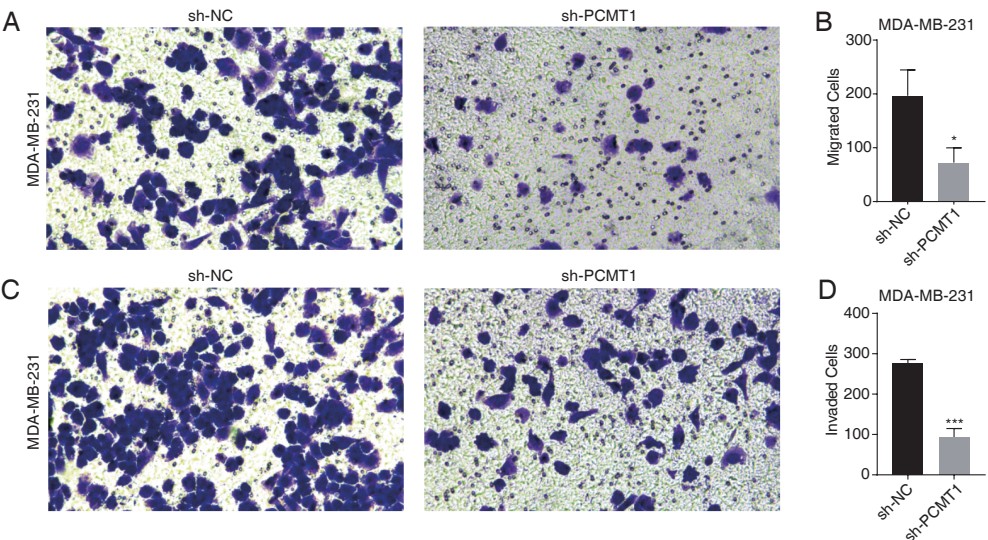

**Figure 2** **PCMT1 knockdown inhibits MDA-MB-231 cell migration and invasion.** (A–B) The cell migration result (A) and quantitative presentation (B) of MDA-MB-231 cells after PCMT1 knockdown. Error bars represent mean ± SEM. $N = 3$. *$p$-value $< 0.05$; Student's $t$-test. (C–D) The cell invasion result (C) and quantitative presentation (D) of MDA-MB-231 cells after PCMT1 knockdown. Error bars represent mean ± SEM. $N = 3$. ***$p$-value $< 0.001$; Student's $t$-test.

migration-associated pathways (Fig. 3E, top panel). The down-DEGs were enriched in cell adhesion-associated pathways (Fig. 3E, bottom panel). The enriched biological process (BP) pathways were consistent with the cellular functions of PCMT1 identified in Fig. 2, suggesting that PCMT1 could modulate the cellular state by regulating the expression levels of associated genes.

Multiple cancer-related DEGs are found to be different between NC and shPCMT1 samples. In total, nine inflammation and cancer-associated DEGs were selected to show their changed expression patterns, comprising four up-DEGs (*ADM*, *FSTL3*, *SPDEF*, and *SPNS2*) and five down-DEGs (*TACSTD2*, *CCL2*, *EDIL3*, *FAM20C*, and *OLFML2A*). To validate the dysregulation of these DEGs, we performed RT-qPCR experiment and found that except *SPDEF*, eight of the nine DEGs showed consistent and significant difference with RNA-seq data (Fig. 3F).

## PCMT1 regulates the expression level of RBPs
As an important protein family, RNA binding proteins (RBPs) play essential roles in transcriptional and post-transcriptional regulation and modulate the progression, expression, and functions of various RNAs (*Gerstberger, Hafner & Tuschl, 2014*; *Hentze et al., 2018*). Thus we analyzed the RBPs dysregulated by shPCTM1 in MDA-MB-231 cells and found 34 RBPs and DEGs overlapped (Fig. 4A). Of the 34 RBPs, 22 were up-regulated and 12 were down-regulated by shPCMT1, and these RBPs changed consistently in the two groups (Fig. 4B). We selected several important cancer and alternative splicing associated RBP genes that were differently expressed by shPCMT1, and validated their expression

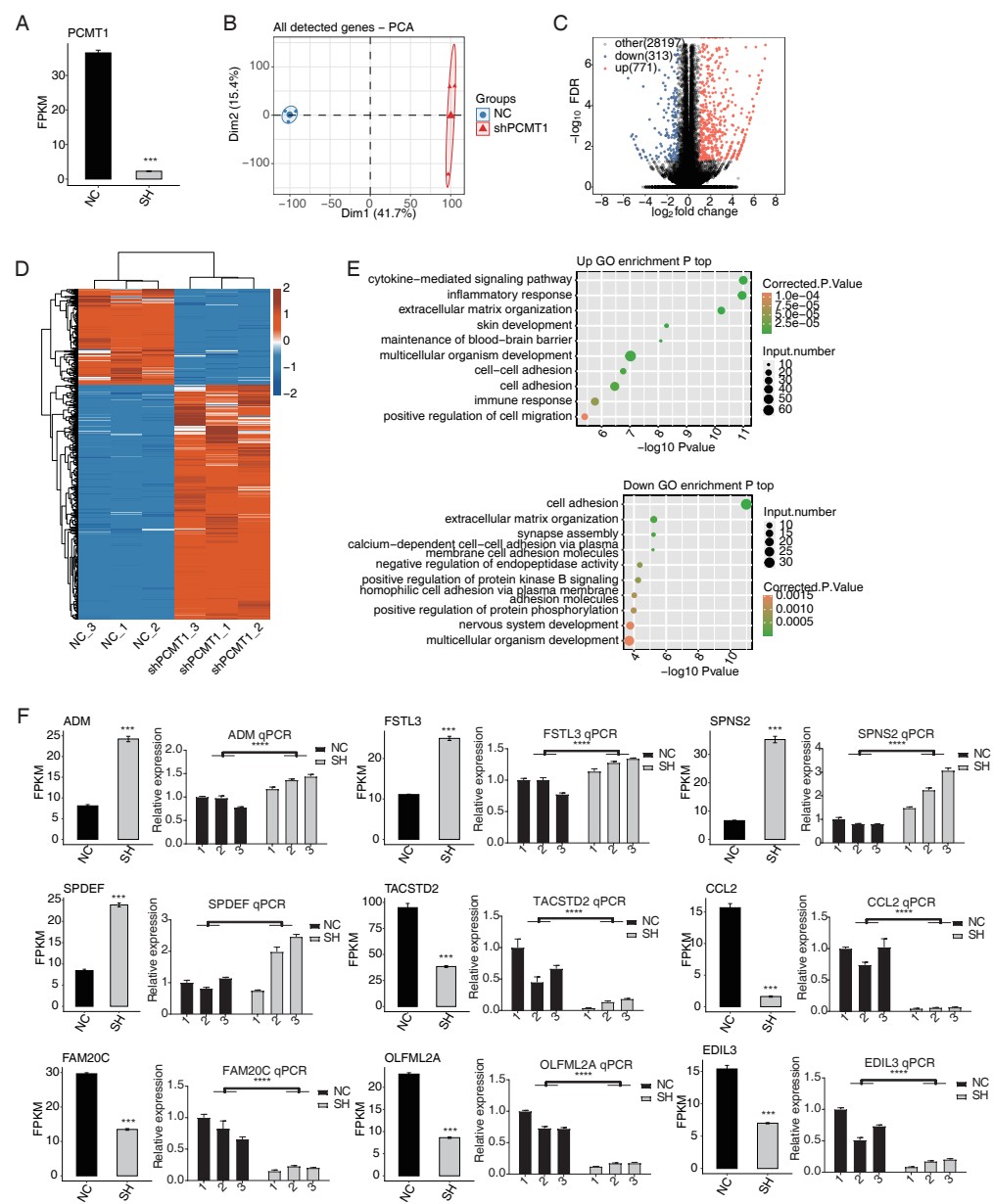

**Figure 3 PCMT1 regulates gene expression in MDA-MB-231 cells.** (A) The bar plot shows the expression pattern and statistical difference of DEGs for PCMT1. Error bars represent mean ± SEM. $N = 3$. ***$p$-value $< 0.001$; Student's $t$-test. (B) PCA result based on FPKM values of all detected genes from sh-PCMT1 and NC groups. The ellipse for each group is the confidence ellipse. (C) The volcano plot displays all DEGs between shPCMT1 and NC samples. (D) The hierarchical cluster heatmap presents the expression levels of all DEGs. (E) The scatter plot shows the top enriched GO BP results of the up-regulated (top panel) and down-regulated (bottom panel) DEGs.The bar plot shows the expression pattern and statistical difference of 9 DEGs by RNA-seq and RT-qPCR validation. Error bars represent mean ± SEM. $N = 3$. ***$p$-value $< 0.001$; ****$p$-value $< 0.0001$; Student's $t$-test.

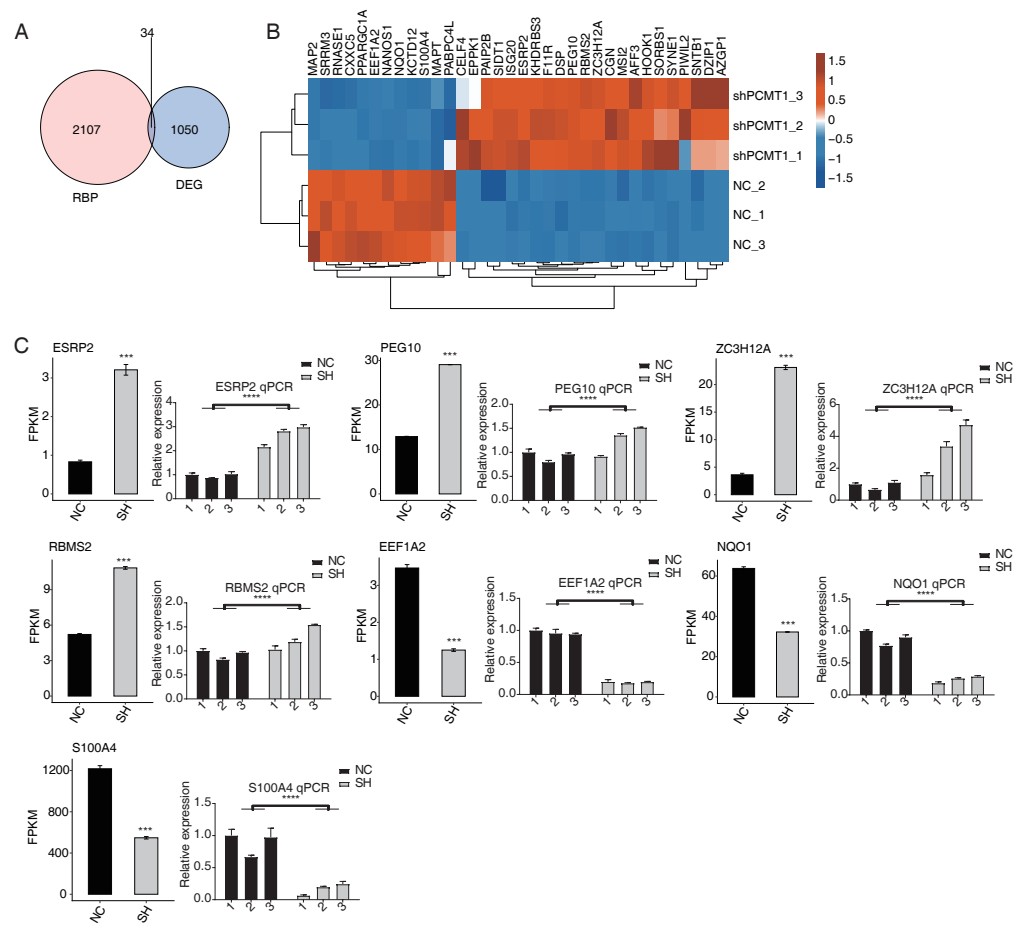

**Figure 4 PCMT1 regulates the expression of RBPs.** (A) Venn diagram shows the overlapped result between RBPs and DEGs. (B) The hierarchical cluster heatmap displays the expression levels of all overlapped RBPs from (A).The bar plot shows the expression pattern and statistical difference of 7 differentially expressed RBPs by RNA-seq and RT-qPCR validation. Error bars represent mean ± SEM. $N = 3$. ***$p$-value < 0.001; ****$p$-value < 0.0001. Student's $t$-test.

pattern by RT-qPCR experiment (Fig. 4C). PCMT1 may regulate these RBPs to influence the AS of cancer-related genes.

## PCMT1 regulates the AS pattern of transcripts in MDA-MB-231 cells

Post-transcriptional regulation, especially AS, plays a critical role in various BPs (*Ule & Blencowe, 2019*). We investigated whether shPCMT1 influenced the AS profile in MDA-MB-231 cells using the RNA-seq data. Using the ABLas program (*Xia et al., 2017*), we identified 2287 PCMT1-regulated ASEs (RASEs) between shPCMT1 and NC samples. These RASEs fell into 9 AS types (Fig. 5A). The ES events increased in shPCMT1 samples, while cassette exon decreased in shPCMT1 samples (Fig. 5A), indicating that PCMT1 knockdown tended to exclude exons during primary RNA processing. According to the GO enrichment analysis of the RASE-embedded genes (RASGs), they were enriched in cancer-related pathways such as cell cycle, cell division, and translation (Fig. 5B).

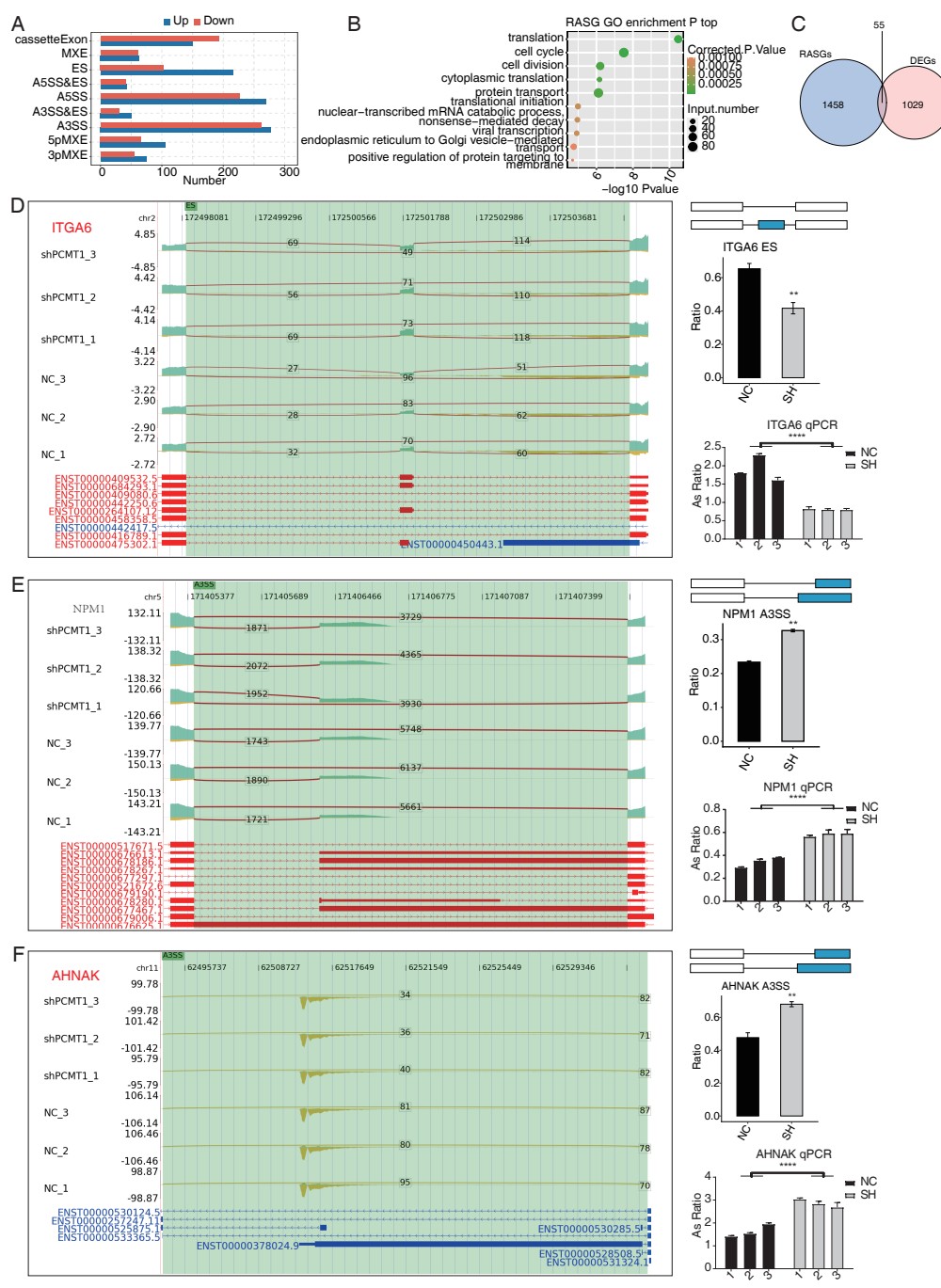

**Figure 5** **PCMT1 regulates the AS of genes in MDA-MB-231 cells.** (A) The bar plot shows the number and types of PCMT1-regulated ASEs. (B) The scatter plot displays the top enriched GO BP pathways of the RASGs. (C) Venn diagram shows the overlapped genes between RASGs and DEGs. (D) PCMT1 regulates the AS of *ITGA6*. Left panel: IGV-sashimi plot presents the RASEs and binding sites across mRNAs. Reads distribution of RASE is shown above and the transcripts of each gene are shown below. (continued on next page...)

**Figure 5 (…continued)**
Right panel: The schematic diagram depicts the structure of ASEs. RNA-seq and RT-qPCR validation of ASEs was shown in the right middle and bottom panel, respectively. (E) The same as (D) but for *NPM1*. Error bars represent mean ± SEM. $N = 3$. **$p$-value $< 0.01$, ****$p$-value $< 0.0001$; Student's $t$-test.The same as (D) but for *AHNAK*. Error bars represent mean ± SEM. $N = 3$. **$p$-value $< 0.01$, ****$p$-value $< 0.0001$; Student's $t$-test.

Meanwhile, we detected several AS pathways and associated genes from RASGs, including *SRSF3*, *LARP7*, *KHDRBS3*, *TIA1*, *HNRNPK*, and *RBM23* (Table S5). To investigate the association between DEGs and RASGs, we found 55 genes overlapped between DEGs and RASGs (Fig. 5C). We selected 10 overlapped RASGs to illustrate the reads distribution, the splicing reads number, and the changed AS ratio of the dysregulated RASEs. They are ES of *ITGA6* (Fig. 5D), A3SS of *NPM1* (Fig. 5E), A3SS of *AHNAK* (Fig. 5F), A5SS of *DDX10* (Fig. S2A), 3pMXE of *ZNF384* (Fig. S2B), A5SS of *DUSP14* (Fig. S2C), cassette exon of *BAIAP2* (Fig. S3A), A3SS of *SNHG29* (Fig. S3B), A3SS of *FOXP1* (Fig. S4A), and A3SS of *CD44* (Fig. S4B). Meanwhile, we validated their changed AS ratios by RT-qPCR experiment and found 8 of the 10 RASEs showed consistent and significant changed AS ratio except FOXP1 and CD44 (Figs. 5D–5F, S2–S4). The expression levels of the genes related to AS are different between NC and shPCMT1 samples, indicating that they may participate in TNBC by regulating the AS of other genes.

# DISCUSSION

The molecular nature of TNBC is an obstacle to its treatment and targeted therapy (*Yin et al., 2020*). The challenges drive us to investigate novel molecular targets and new therapeutic strategies. PCMT1 is a protein methyltransferase enzyme that is highly expressed in multiple cancers and plays an important role in cancer development. A recent study demonstrated the cellular functions and upstream regulatory axis of PCMT1 in breast cancer cell lines (*Zhang et al., 2022a*; *Zhang et al., 2022b*). While the molecular targets of PCMT1 in breast cancer remain unknown. In this study, we silenced PCMT1 in MDA-MB-231 cells and explored its downstream molecular targets to decipher the underlying mechanisms by which PCMT1 promotes the progression of breast cancer. We found the expression levels of cell adhesion- and inflammation-associated genes were dysregulated due to shPCMT1,and the AS pattern of cell cycle-associated genes was also dysregulated, indicating that PCMT1 could induce the malignancy of breast cancer cells by modulating the transcriptome profiles of these genes.

The cellular transcriptome profile is regulated by multiple proteins in a direct or indirect interaction manner. In this study, we identified 1084 genes whose expression levels were regulated by PCMT1 to investigate how PCMT1 regulates the transcriptome. Meanwhile, we found that the dysregulated DEGs were highly associated with the cellular functions of PCMT1. The top enriched cell adhesion-related pathways participate in cell migration and invasion (*Jacquemet, Hamidi & Ivaska, 2015*). We found the cell adhesion-associated genes were both up- and down-regulated by PCMT1, indicating that the cellular junction proteins were dysregulated among cells. This dysregulation may inhibit the migration and invasion
by shPCMT1. Meanwhile, we also found that the immune/inflammation-associated genes were up-regulated by shPCMT1. A previous study demonstrated that tumor tissues from TNBC patients included mixed and compartmentalized cells, and these cells were associated with compartmentalization and survival (*Keren et al., 2018*). In TNBC treatment, there is a potential for immunotherapy-based combination strategies to use immune checkpoint inhibitors to enhance both innate and adaptive immune responses (*Thomas, Al-Khadairi & Decock, 2020*). These findings together indicate that PCMT1 may be a possible therapeutic target for TNBC immunotherapy in the future.

Among the DEGs regulated by PCMT1, several DEGs are associated with TNBC. For example, *SPNS2* encodes GUSBP11 which regulates TNBC progression by modulating the miR-579-3p/SPNS2 axis (*Wu, Sun & Qin, 2022*). *FSTL3* expression is inversely associated with tumor size and nuclear grade in invasive breast cancer (*Couto et al., 2017*). *FST* is proven to be a bona fide metastasis suppressor (*Seachrist et al., 2017*). *ADM* is a valuable biomarker for TNBC prognosis and an anti-metastasis candidate therapeutic target in TNBC (*Liu et al., 2020*). *SPDEF* expression inhibits cell migration and invasion partially by down-regulating EMT-related protein markers, indicating that it plays a critical role in the suppression of TNBC cell metastasis (*Yousefi et al., 2021*). We detected 34 RBPs from the DEGs and validated their expression pattern of 7 RBPs using RT-qPCR. These RBPs could regulate TNBC progression. For example, NQO1 helped reduce the resistance to TNBC therapies (*Cao et al., 2014*). S100A4 may have stimulatory effects elicited in TNBC cells through FGF2/FGFR1 signaling pathway (*Santolla, Talia & Maggiolini, 2021*). RBPs have a wide range of effects on the RNAs from synthesis to degradation, influence every step in the progression of cancer, and are emerging as a therapeutic target for cancer prevention and treatment (*Hong, 2017*; *Chen, Qin & Zheng, 2022*). We propose that these dysregulated RBPs participate in the progression of TNBC and that PCMT1 regulates the cellular states and transcriptome profile perhaps by modulating the expression of these RBPs. Our propositions need to be further investigated in future studies.

Another interesting finding is that PCMT1 modulates the AS pattern of primary transcripts, which has not been reported. We speculate that PCMT1 may modulate the global AS profile by regulating the functions of spliceosome-associated proteins, several of which were detected from the RASGs (Table S5). The ES events were specifically up-regulated in shPCMT1 samples, indicating that PCMT1 could affect splicing proteins involved in ES. A previous study reported that PCMT1 protein contains RNA binding activity and methyltransferase domain; the silencing of PCMT1 in HeLa cells affects the cell cycle and increases cells in G2/M phases (*Enunlu et al., 2003*), consistent with the finding that RASGs by PCMT1 were enriched in cell cycle pathways, which is also a possible explaintion for the inhibited cell proliferation by shPCMT1. Based on the RNA methylation function of PCMT1, we also speculate that PCMT1 regulate AS of target RNAs probably by methylating RNA targets. We validated the changed splicing pattern of several important RASGs. *ITGA6* is found to be differently expressed in 2 sample groups. *ITGA6* is potentially involved in the regulation of actin cytoskeleton, thus playing a role in TNBC migration (*Klahan et al., 2014*). *ITGA6* is a relevant regulatory target for the treatment of cisplatin activity (*Cataldo et al., 2020*). Another DEG *FOXP1* is a transcriptional

regulator of lymphocyte development and is aberrantly expressed in some human tumors (*De Silva et al., 2019*). It may serve as a winged helix/forkhead TF (*Chiang et al., 2017*). *FOXP1* might predict poor overall survival (OS) in patients with specific cancer types (*Yu et al., 2018*). DEG *AHNAK* has been considered to act as a tumor suppressor that negatively regulates TNBC cell proliferation (*Chen et al., 2017*). And it has been proven to play a role in the chemotherapeutic response to breast cancer cells (*Davis et al., 2018*). Studies are needed to further explore how PCMT1 regulates these ASEs by cooperating with other proteins and how these RASEs regulate TNBC.

In summary, we identified the cellular functions and molecular targets of PCMT1 in TNBC MDA-MB-231 cells, expanding our understanding of the functions of PCMT1 in breast cancer. We found that PCMT1 has an impact on cell proliferation, migration, and invasion by regulating transcriptome profiles. These results suggest a novel functional mechanism for PCMT1 in TNBC progression by modulating the RNA expression and splicing patterns. The identified molecular targets of PCMT1 could serve as potential therapeutic targets for future TNBC treatment. Meanwhile, we have realized the limitations of this study, including the single cell line used in this study, the shortage of directly functional evidence between PCMT1 and target genes, and the absence of clinical samples and animal model validation. Thus, more *in vitro* and *in vivo* experiments, including the mutant cells with different levels of PCMT1 knockdown, and further studies are needed to validate the potential targets regulated by PCMT1.

### Funding
The authors received no funding for this work.

### Competing Interests
Xinyu Hu and Dong Chen are employed by Wuhan Ruixing Biotechnology Co., Ltd.. The authors declare there are no competing interests.

### Author Contributions
- Aili Saiding performed the experiments, prepared figures and/or tables, authored or reviewed drafts of the article, and approved the final draft.
- Dilinuer Maimaitiyiming performed the experiments, prepared figures and/or tables, authored or reviewed drafts of the article, and approved the final draft.
- Minglan Chen performed the experiments, authored or reviewed drafts of the article, and approved the final draft.
- Futian Yan performed the experiments, authored or reviewed drafts of the article, and approved the final draft.
- Dong Chen analyzed the data, authored or reviewed drafts of the article, and approved the final draft.
- Xinyu Hu analyzed the data, prepared figures and/or tables, authored or reviewed drafts of the article, and approved the final draft.

- Ping Shi conceived and designed the experiments, authored or reviewed drafts of the article, and approved the final draft.

## Data Availability

The data are available at NCBI GEO: GSE225762.

## Supplemental Information

Supplemental information for this article can be found online at http://dx.doi.org/10.7717/peerj.16006#supplemental-information.

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
