# Peer review of "PCMT1 knockdown attenuates malignant properties by globally regulating transcriptome profiles in triple-negative breast cancer cells"

_PeerJ, doi:10.7717/peerj.16006_

## Round 0.1 · original submission · Minor Revisions

Dear authors,

Thank you for revising and resubmitting your previously rejected manuscript. After careful consideration by two experts in the field, I kindly request you attend to minor considerations and suggestions made by the reviewers. I think these suggestions are relevant and can be attended to swiftly.

Thank you for resubmitting this work to PeerJ.

Best regards,
Bernardo

Reviewer 1 ·

Basic reporting

Overall, the authors’ presentation of their findings are well organized, well described, and support their conclusions. Raw data is provided as additional evidence, and individually identified targets are further evaluated to confirm preliminary results. Targets chosen for evaluation show a balanced approach, as not all in depth assessments support the initial findings, however, this does not invalidate the authors’ approach or general conclusions. The language is clear and succinct.

One area of reporting that could strengthen the manuscript would be to include greater experimental detail in the figure legends, as seemed to be the case in the original manuscript but were edited out in the final submitted version. Many readers will peruse the figures to gain a sense of the study’s findings, but with the abbreviated figure legends in the final version, it is difficult to gain a clear understanding of each of the figures without referring to the text of the manuscript.

Experimental design

The authors use a robust approach to ask their scientific questions. The study is well defined and well controlled. The results represent an important foundation for further inquiry regarding identifying novel therapeutic targets to increase treatment options and efficacy for women with triple negative breast cancer. Conclusions are well supported by the data and are not overstated.

Validity of the findings

The authors take multiple approaches to validate their findings, including identifying individual targets for further quantification to support initial screening results. The impact of the findings is limited by the fact that a cell line, rather than primary tissue from patients with triple negative breast cancer was evaluated, but the authors do state this limitation to their study and encourage similar analyses be performed in other cell lines and primary tissue.

Additional comments

1. Adding greater experimental detail to the figure legends throughout the manuscript will add clarity to the figures and enhance the overall “messaging” of the manuscript.
2. The authors present proliferation data in Figure 1D. It appears that the rate of proliferation is recovering with time. Does the rate of proliferation of MDA-MB-231 cells transfected with shPCMT1 recover at later time points? i.e. at 96 hours or beyond?
3. The authors suggest that differences in apoptosis is the reason for differing proliferation rates, but the data presented in Fig. S1B-C do not support this conclusion. Can the authors provide additional discussion as to what might be influencing the differences in proliferation rates observed at the earlier timepoints?
4. The difference between the “migration” and “invasion” assays appears only to rest with the number of cells initially seeded in the upper transwells. Can the authors clarify how these two assays are differentially used to assess migration vs. invasion?

Reviewer 2 ·

Basic reporting

This study was resubmitted to this journal. The authors has revised the manuscript according to the comments. However, there are still some additional flaws.

Experimental design

no comment

Validity of the findings

no comment

Additional comments

This study was resubmitted to this journal. The authors has revised the manuscript according to the comments. However, there are still some additional flaws.

1.In the “Material and Methods” section, you should introduce RT-qPCR.
2.As the authors did not perform experiment to study the relationship between PCMT1 and the identified molecular targets, you should expand the discussion.
(1)This study speculates that PCMT1 may modulate the global AS profile by regulating the functions of spliceosome-associated proteins. Is there any spliceosome-associated protein in DEGs? You could discuss the possible relationship between PCMT1 and some splice proteins.
(2)You mention that PCMT1 protein contains RNA binding activity and methyltransferase domain. Did PCMT1 methylate mRNA of target genes to regulate the expression of targets gene?

---

## Round 0.2 · accepted · Accept

Dear authors,

It is my pleasure to announce that the revised version of this paper has fulfilled all the comments and revisions done by the expert reviewers.

I thank the authors for choosing PeerJ and the reviewers for the positive insight.

All the best for your research moving forwards and congratulations.

Best regards,
Bernardo